# Analysis and Verification of Heat Dissipation Structures Embedded in Substrates in Power Chips Based on Square Frustums Thermal through Silicon Vias

**DOI:** 10.3390/mi15030323

**Published:** 2024-02-26

**Authors:** Fengjie Guo, Kui Ma, Jingyang Ran, Fashun Yang

**Affiliations:** 1Department of Electronics, Guizhou University, Guiyang 550025, China; guofengjie000000@163.com (F.G.); kma@gzu.edu.cn (K.M.); jingyang0016@126.com (J.R.); 2Reliability Engineering Research Center of Semiconductor Power Devices of Ministry of Education, Guiyang 550025, China; 3Guizhou Provincial Key Lab of Micro-Nano-Electronics and Software Technology, Guiyang 550025, China

**Keywords:** thermal reliability, thermal characteristics, dissipative structure, square frustums TTSV

## Abstract

A novel heat dissipation structure composed of square frustums thermal through silicon via array and embedded in P-type (100) silicon substrate is proposed to improve the heat dissipation capacity of power chips while reducing process difficulty. Based on theoretical analysis, the heat transfer model and thermo-electric coupling reliability model of a power chip with the proposed heat dissipation structure are established. A comparative study of simulation indicates that the proposed heat dissipation structure, which can avoid problems such as softness, poor rigidity, fragility and easy fracture caused by thinning chips has better heat dissipation capability than thinning the substrate of power chips.

## 1. Introduction

Power semiconductor chips include power devices and power integrated circuits. With the improvement of integration, power chips tend to have high power density. They are widely used in the fields of new energy power generation, aerospace, high-speed locomotive traction, hybrid electric vehicles and industrial motor drivers with high-reliability demand. Reliability is the most important requirement of power semiconductor chips [1]. Practice has proved that the main causes of power electronic components failure are temperature, humidity, vibration and dust. Temperature accounts for more than half of these four factors and is the main cause of electronic components’ failures [2]. Research shows that when the temperature of the device increases by 10 K, the life failure rate of the device doubles. Additionally, increased temperatures result in quality deterioration and degraded performance with thermal throttling [3]. Therefore, thermal management is important in order to increase the lifetime and protect performance.

The research on thermal management has focused on reducing thermal resistance in three parts. The first method is to use external cooling strategies to reduce the thermal resistance from the device to the atmosphere, such as heat pipe [4] micro-channel coolers, piezoelectric fans [5] and thermoelectric coolers [6]. This kind of cooling is low cost and the temperature can be reduced quickly; however, there are many problems, such as poor circulation and uneven heat dissipation [7,8,9]. The second method is to reduce the thermal resistance of packaging structures, such as by using new substrate material, solder [10], etc. The third method is to reduce the thermal resistance of the die itself, such as via a wafer thinning process [11]. In 1989, Koyanagi et al. of Tohoku University in Japan proposed a process for manufacturing 3D integrated circuits for the first time. After bonding the wafer to another thick wafer, it is ground thin from the back of the wafer. However, in practical application, the wafer is too thin to be subjected to a large thermal gradient, and the heat generated by a single device cannot be diffused horizontally, resulting in the generation of hot spots [12]. Moreover, wafer thinning has thickness limitations. The silicon substrate must have a certain thickness (200 μm~300 μm) in the silicon body area to provide mechanical support and the chemical mechanical polishing (CMP) process of preparing thinner (<200 μm) chips is more difficult. Embedded packaging is currently the most advanced idea [13]. Various types of through silicon vias (TSVs) such as cylindrical TSVs, annular TSVs, square frustum TSVs, etc., are widely used for signal connection and heat dissipation. Vertical interconnection between upper and lower chips has the advantages of small size, low power consumption, high interconnection density and heterogeneous integration. Similarly, through silicon vias can also reduce their own thermal resistance [14,15,16]. It can not only be used as a channel for signals in 3D integrated circuits but also as a channel for heat dissipation in 3D integrated circuits [17].

In this paper, a low-cost heat dissipation structure embedded in a substrate of power chips is studied. Square frustum thermal through silicon vias (SF-TTSVs), which can be formed by KOH wet corrosion for etching vias, sputtering for titanium barrier layer and copper seed layer preparation, and copper electroplating for vias filling, are embedded in P-type (100) silicon bulk regions of power semiconductor chips as heat dissipation channels. The SF-TTSVs inside a power semiconductor chip do not affect the layout of the device region and high conductivity and high thermal conductivity materials refilled in SF-TTSVs can greatly improve the conductivity and thermal conductivity of the bulk region in a power semiconductor chip. It is significant to improve the heat dissipation performance of high-power chips at a low cost [18]. The embedded heat dissipation structure is designed to improve the thermal reliability of power dies, and the feasibility of the design is verified by *COMSOL Multiphysics* 6.2. Compared with thinned chips, SF-TTSVs can almost penetrate the silicon substrate, the capability of heat conduction of the proposed structure is better, and the fabrication of the proposed structure is simpler.

Through silicon via play a vital role in enabling advanced integrated systems, but their development is greatly hindered by multiphysics coupling effects. The multiphysics field coupling process of TSV is very complicated, and the thermal field distribution, electromagnetic field distribution and structural distribution are related and interact. Aiming at the multiphysics field coupling problem of TSV using *COMSOL Multiphysics* software for modeling can give intuitive results. The influence of TSV physical structure size parameters (radius, aspect ratio, insulating layer thickness and TSV filling) on the thermal conduction of TSV and the advantages and disadvantages of different parameters can be clearly seen in the simulation results [19,20,21,22]. Therefore, this paper uses *COMSOL Multiphysics* to establish a simulation model to verify the effectiveness of the embedded heat dissipation structure.

## 2. Structure Design and Theoretical Analysis

### 2.1. Design of the Embedded Heat Dissipation Structure

A semiconductor power chip can be divided into device region and bulk region. As shown in Figure 1, SF-TTSVs are embedded into the bulk region to form an array. It is effective at improving heat dissipation capabilities of refilled high conductivity and high thermal conductivity materials such as copper, silver, gold and carbon nanotube, etc., into the SF-TTSVs. Compared to other materials, copper has a lower cost. Many studies on copper-filled TSVs have been reported [15,16,17,18,19,20,21,22,23]. A barrier layer that is used to block copper diffusion into silicon is needed for filling copper in the vias. Titanium, tantalum, titanium nitride, tantalum nitride and soft organic compounds, etc., can be used as barrier layers in TSVs. Many studies in the literature have reported that covering the inner walls of TSVs with an extremely thin layer of titanium (~100 nm) can block the diffusion of copper into silicon. Even though using soft organic compounds as the barrier layer is effective at reducing mechanical stress, a thicker soft organic compound barrier layer will compress the volume of copper filling in the TSV, thereby reducing the heat dissipation efficiency of the proposed heat dissipation structure. Therefore, we selected titanium as the barrier layer and copper as the filling material. Because the titanium barrier layer is extremely thin, the thermal resistance of this layer is ignored. Meanwhile, refilled high-conductivity materials will reduce parasitic resistance of the bulk area and the heat generated by large current flowing through the bulk region will be reduced. In this structure, the SF-TTSV array does not pass through the device region where devices or circuits are located, so it is not necessary to consider the effects of thermal stress resulting from SF-TTSVs on the device region. Therefore, the device region is regarded as an integral heat source. 

Interventionary segmented thermal resistance model of an SF-TTSV in the heat dissipation array as shown in Figure 1, is established in Figure 2. According to Kirchhoff’s law, we can write the following equations:(1)$$\frac{T}{\left(\frac{1}{R_{t3}}+\frac{1}{R_{t4}}\right)}=q$$
(2)$$\frac{T}{R_{t1}+R_{t2}}=q$$
where *T* is the temperature, *q* is the heat generated by the device region of the chip, *R*_t1_ is the thermal resistance of the device region, *R*_t2_ and *R*_t4_ are the thermal resistance of the bulk region above the SF-TTSV and the thermal resistance of the bulk region on the right and left sides of the SF-TTSV, respectively, and *R*_t3_ is the vertical thermal resistance of the SF-TTSV. 

According to Fourier’s law, we can get,
(3)$$q=kA\frac{△T}{L}$$
where *L* is the length of the heat path through the material, *k* is the thermal conductivity of the material, *A* is the cross-sectional area of the heat path through the material and Δ*T* is the temperature rise.

The thermal resistance, *R*, can be obtained by analogy with Ohm’s law:(4)$$R=\frac{L}{kA}$$

It can be seen from Equation (4) that thermal resistance is inversely proportional to the thermal conductivity of the material. Increasing the thermal conductivity of the material in the heat dissipation path is beneficial to improving heat dissipation efficiency [23].

In Figure 3a, the thermal resistance of a cell with half the SF-TTSV contains the thermal resistance of the device region, the silicon bulk region, and half of the SF-TTSV. Assume the total thermal resistance is *r*_1_. In Figure 3b, thermal resistance of a cell without SF-TTSV under the same size as in Figure 3a contains the thermal resistance of the device region and the silicon bulk region. Assume the total thermal resistance is *r*_2_.
(5)$$r_{1}=R_{t1}+R_{t2}+R_{t3}//R_{t4}$$
(6)$$r_{2}=R_{t1}+R_{t2}+R_{t5}$$

*R*_t1_ and *R*_t2_ are the same for Figure 3a,b. SF-TTSV refilled high thermal conductivity materials make *R*_t3_ // *R*_t4_ < *R*_t5_. So, *r*_1_ < *r*_2_. The symbol // indicates that resistors are connected in parallel. 

In an embedded heat dissipation structure as shown in Figure 1, the volume ratio of the refilled high-conductivity material in the bulk region is not high. The 54.7° tilt angle of the SF-TTSV results in a beveled shape in the bulk region of the chip, accounting for a relatively small volume of the entire chip substrate. Only one SF-TTSV embedded in the whole bulk region of a power chip, as shown in Figure 4, can increase the volume ratio of the refilled material.

### 2.2. Theoretical Analysis

In Figure 5, the axial direction is z and the radial directions are x and y. The bulk region of a power chip is divided into part a and part b.

Based on the one-dimensional Fourier heat conduction law, the z-direction equivalent heat fluxes of SF-TTSV and silicon of part b are *q*_T_ and *q*_Si_, respectively.
(7)$$q_{T}=\frac{k_{T}\times \Delta T}{H_{1}}$$
(8)$$q_{Si}=\frac{k_{Si}\times \Delta T}{H_{1}}$$

*k*_T_ and *k*_Si_ are the thermal conductivities of SF-TTSV and silicon, respectively, as the temperature rises.

In part b, the equivalent heat flux in the z-direction is
(9)$$q_{z}=\frac{k_{z}\times \Delta T}{H_{1}}$$

*k*_z_ is the equivalent thermal conductivity in the z-direction of part b.

Based on the law of conservation of energy, the total heat flux is
(10)$$Q=\frac{q_{T}V_{T}}{H_{1}}+\frac{q_{Si}V_{Si}}{H_{1}}$$

It can be deduced that the thermal conductivity of the equivalent block is
(11)$$k_{z}=\frac{k_{T}\times {\;V}_{T}+{\;k}_{Si}\times V_{Si}}{V_{T}+V_{si}}$$

*V*_T_ is the volume of SF-TTSV and *V*_Si_ is the volume of silicon. *V* = *V*_T_ + *V*_Si_ is the whole volume in part b.

The volume ratio of SF-TTSV to silicon in part b is
(12)$$\alpha =\frac{V_{T}}{V_{Si}}$$
(13)$$k_{z}=\frac{k_{T}\times \alpha +{\;k}_{Si}}{1+\alpha }$$

The thermal resistance of part b is
(14)$$R_{tb}=\frac{H_{1}}{k_{z}A}$$

*A* is the cross-sectional area perpendicular to the z-direction

The thermal resistance of part a is
(15)$$R_{ta}=\frac{H_{0}-H_{1}}{k_{Si}A}$$

The total thermal resistance of the bulk region is
(16)$$R_{t}=R_{ta}+R_{tb}$$

Similarly, assuming the conductivity of SF-TTSV is *σ*_T_ and the conductivity of silicon is *σ*_Si_, the equivalent conductivity in the z-direction of part b in Figure 5 is
(17)$$\sigma_{z}=\frac{\sigma_{T}\times \alpha +\sigma_{Si}}{1+\alpha }$$

The resistance of part b is
(18)$$R_{eb}=\frac{H_{1}}{\sigma_{z}A}$$

The resistance of part a is
(19)$$R_{ea}=\frac{H_{0}-H_{1}}{\sigma_{Si}A}$$

The total resistance of the bulk region is
(20)$$R_{e}=R_{ea}+R_{eb}$$

When current *I* flows through the bulk region, the dissipated power is
(21)$$Q_{s}=I^{2}R_{e}$$

Under thermo-electric coupling, the temperature rise Δ*T* of the power chip can be calculated as
(22)$$\Delta T={(Q}_{s}{+P)R}_{t}$$
where *P* is the power of the device region.

The maximum temperature T of the chip is
(23)$$T=T_{a}+Q_{s}R_{t}=T_{a}+I^{2}R_{e}R_{t}+PR_{t}$$

*T*a is the ambient temperature.

Taking the case of refilling copper into SF-TTSV as an example, setting the chip area to 5 × 5 mm^2^, *H*_0_ = 625 μm, *H*_1_ = 500 μm, *R*_bottom_ of SF-TTSV is 4.9 mm, current *I* = 10 A, conductivity of SF-TTSV, *σ*_T_ = 5.998 × 10^7^ S/m, the conductivity of silicon *σ*_Si_ = 125 S/m, thermal conductivity of SF-TTSV and heat sink $k_{T}=400\;W/(m\cdot K)$,the thermal conductivity of silicon $k_{Si}=130\;W/(m\cdot K)$, the ambient temperature is 300 K.

The parameter values are shown in Table 1.

A model with the same parameters is established in COMSOL Multiphysics. As shown in Figure 6, the maximum temperature is 314.066 K. In the thermal distribution map generated by the *COMSOL Multiphysics* software, the upper temperature scale is the temperature range in the vertical direction and the lower temperature scale is the temperature range in the horizontal direction of the upper surface. The difference in the maximum temperature between calculated results and simulated results is 0.43%. Therefore, the theoretical calculation is proven to be believable.

## 3. Validation of Simulation Analysis

For power chips, parasitic thermal resistance and parasitic resistance of the silicon bulk region are usually reduced by a thinning wafer. It is easy to damage the wafer during the thinning process, and the silicon bulk region must have a certain thickness (200 μm~300 μm) for mechanical support. Thus, the proposed heat dissipation structure is simulated in this paper and compared with thinned power chips to verify its feasibility and effectiveness.

At present, the substrate in most integrated circuit chips is the P-type monocrystalline silicon. Some power devices, such as p-channel MOSFETs, p-channel IGBTs, etc., also have a P-type substrate. P-type (100) monocrystalline silicon can be etched to form square frustums using an aqueous KOH solution. For integrated circuits, the resistance of the P-type substrate is ~10 Ω·cm. For power devices, the resistance of the P-type substrate is 10^−2^~10^−3^ Ω·cm. 

In power chips, the thermal resistance of the bulk region directly affects heat transfer characteristics. Parasitic resistance of the bulk region can cause additional heat because of the large current flows through this region. The heat generated in the device region flows through the bulk region, superimposed with the heat generated in the bulk region, and then flows into the heat sink.

### 3.1. Thermoeletronic Simulation Analysis

Simulation models of power chips with or without SF-TTSVs were established in *COMSOL Multiphysics*, as shown in Figure 7. The width and length of the power chip were set to 3.2 mm, the thickness of the device region was set to 14 μm, the thickness of the bulk region *(H*_0_) was 600 μm, the material filling the SF-TTSVs was copper, *R*_bottom_ of SF-TTSV was 880 μm, the height of SF-TTSV (*H*_1_) was 500 μm, the spacing between adjacent SF-TTSVs was 140 μm and the size of the heat sink was 3.2 mm × 3.2 mm × 5 mm.

Assuming the current density of the power chips is 50 A/cm^2^, the parasitic resistance of the device region is 140 mΩ and the resistance of the P-type (100) monocrystalline silicon substrate is 8 Ω·cm. According to the equations in Section 2, the resistance of power chips with SF-TTSV array, with single SF-TTSV, and with thinned bulk region is calculated in Table 2. Maximum conduction current (Mcc) is equal to current density times chip area. For different *H*_1_, *R*_bottom_ is fixed at 880 μm and the spacing between adjacent SF-TTSVs is fixed at 140 μm.

In the simulation model, copper is refilled into the SF-TTSVs. There is a barrier layer between the bulk silicon and the refilled copper. Because this layer is ultra-thin (~100 nm), its influence on heat transfer can be ignored. The thermal power of the bulk region is $P=I^{2}R$. Where *I* is the current flow through the bulk region and *R* is the parasitic resistance of the bulk region. The temperature of the bottom surface of the copper heat sink (ambient temperature) is set to room temperature (300 K). It is assumed that heat can be completely transferred into the atmosphere through the bottom surface of the heat sink and other surfaces of the heat sink are adiabatic.

Firstly, the thermal distribution of a power chip without SF-TTSV and without thinning is simulated. The current flow through the 3.2 mm × 3.2 mm power chip is set to 1 A. Figure 8a shows that the temperature rise in the chip is 7.821 K under heat transfer conditions while under thermoelectric coupling conditions, the temperature rise in the power chip is 8.493 K, as shown in Figure 8b. It is indicated that @ 1 A, the maximum temperature of the chip increases by 0.672 K. Joule heat generated by the parasitic resistance of the silicon bulk region induces an extra temperature rise in the power chip.

Secondly, the thermal distribution of a power chip with an SF-TTSV array is simulated. The current flow through the 3.2 mm × 3.2 mm power chip with the SF-TTSV array in the bulk region is also set to 1 A. Figure 9a shows that the temperature rise in the chip is 1.426 K under heat transfer conditions while under thermoelectric coupling conditions, the temperature rise in the power chip is 1.734 K, as shown in Figure 9b. The maximum temperature in the power chip increases by 0.308 K, considering Joule heat generated by the parasitic resistance of the bulk region @ 1 A.

Because of the high thermal conductivity of copper refill in SF-TTSVs, the thermal resistance of the bulk region in the power chip with SF-TTSV array is reduced compared with the power chip without SF-TTSV array and without thinning. Figure 8a and Figure 9a indicate that under heat transfer conditions, the temperature of the power chip with the SF-TTSV array cooling structure is 6.395 K lower than that of the power chip without the SF-TTSV array and without thinning. The copper refilled in SF-TTSVs also reduces the parasitic resistance of the bulk region. Thus, under thermoelectric coupling conditions, the temperature of the power chip with the SF-TTSV array cooling structure is 6.759 K lower than that of the power chip without the SF-TTSV array and without thinning. The SF-TTSV array cooling structure causes an obvious temperature drop and the temperature drop under thermoelectric coupling conditions is more obvious than that under heat transfer conditions.

### 3.2. Simulation and Comparative Analysis of Power Chips with SF-TTSV Array and Thinned Power Chips

The proposed heat dissipation structure, in which the SF-TTSV array is embedded into the bulk region of the power chip, and the thinned power chip structure are simulated under thermoelectric coupling conditions.

The parasitic resistance of the device region is set to 140 mΩ, the current density is set to 50 A/cm^2^, the chip area is set to 3.2 mm × 3.2 mm as an example and the thickness of the device region is 14 μm. Thermal resistance and parasitic resistance of different structures can be calculated using the equations in Section 2 and the maximum temperature at a certain current can also be calculated.

At different currents, the maximum temperature of power chips with SF-TTSV array @ *H*_0_ = 600 μm, *H*_1_ = 500 μm and 550 μm, respectively, and thinned power chips @ *H*_0_ = 100 μm, 150 μm, 200 μm, 250 μm, 300 μm, 350 μm, 400 μm, respectively is simulated. As shown in Figure 10, temperature of thinned power chips @ *H*_0_ = 100 μm, 150 μm, 200 μm, 250 μm, 300 μm, 350 μm and 400 μm increase from 300 K to 334.72 K, 350.06 K, 366.94 K, 384.57 K, 403 K, 422.24 K and 438.73 K, respectively, when the current increases from 0 to 5.12 A. For thinned power chips, the higher the *H*_0_, the more the maximum temperature increases with the increase in the current. Temperatures of power chips with SF-TTSV array @ *H*_0_ = 600 μm, *H*_1_ = 500 μm and 550 μm are 347.29 K and 330.02 K, respectively, when the current increases from 0 to 5.12 A. The maximum temperature of power chips with SF-TTSV array @ *H*_0_ = 600 μm, *H*_1_ = 500 μm and 550 μm is lower than that of thinned power chips @ *H*_0_ = 200 μm, 250 μm, 300 μm, 350 μm and 400 μm, within 0 A~5.12 A current range. The curve of thinned power chip @ *H*_0_ = 100 μm is bounded by curves of power chips with SF-TTSV array @ *H*_0_ = 600 μm, *H*_1_ = 500 μm and *H*_1_ = 550 μm on the temperature axis. The maximum temperature of power chips with SF-TTSV array @ *H*_0_ = 600 μm, *H*_1_ = 500 μm and 550 μm is close to that of thinned power chip @ *H*_0_ = 100 μm and 150 μm, when the current is lower than 2 A. Once the current exceeds 2 A, at the same current, the maximum temperature from high to low corresponds to thinned power chip @ *H*_0_ = 150 μm, power chip with SF-TTSV array @ *H*_0_ = 600 μm, *H*_1_ = 500 μm, thinned power chip @ *H*_0_ = 100 μm, and power chip with SF-TTSV array @ *H*_0_ = 600 μm, *H*_1_ = 550 μm. Heat dissipation capabilities of power chips with an SF-TTSV array are significantly better than those of thinned power chips. Heat dissipation performance of a power chip with SF-TTSV array @ *H*_0_ = 600 μm, *H*_1_ = 500 μm is better than that of a thinned power chip @ *H*_0_ = 150 μm. Those of power chips with SF-TTSV array @ *H*_0_ = 600 μm and *H*_1_ = 550 μm are even better than those of a thinned power chip @ *H*_0_ = 100 μm. At the maximum current of 5.12 A, the maximum temperature of power chips with SF-TTSV array @ *H_0_* = 600 μm and *H*_1_ = 550 μm is 4.7 K lower than that of a thinned power chip @ *H*_0_ = 100 μm. At the maximum current of 5.12 A, the maximum temperatures of the power chip with SF-TTSV array @ *H*_0_ = 600 μm and *H*_1_ = 500 μm reduced by 0.8%, 5.36%, 9.69%, 13.82%, 17.75% and 20.84% compared to the thinned power chips @ *H*_0_ = 150 μm, 200 μm, 250 μm, 300 μm, 350 μm and 400 μm, respectively, while the maximum temperature of the power chip with SF-TTSV array @ *H*_0_ = 600 μm and *H*_1_ = 550 μm reduced by 1.4%, 5.72%, 10.06%, 14.18%, 18.11%, 21.84% and 24.78% compared to the thinned power chips @ *H*_0_ = 100 μm, 150 μm, 200 μm, 250 μm, 300 μm, 350 μm and 400 μm, respectively. Even though there are slight changes in material properties and geometric parameters, it can be inferred that the power chip with SF-TTSV array @ *H*_0_ = 600 μm and *H*_1_ = 550 μm has better heat dissipation capability than the thinned power chips @ *H*_0_ = 200 μm. Under high current conditions, the advantage of the heat dissipation capacity of power chips with an SF-TTSV array is more obvious.

The curves in Figure 10 indicate that the temperature of chips increases with the increase in the current. The stronger the heat dissipation capacity of the chip, the smaller the rate of change in the curve with the current. Therefore, thinner power chips and power chips with the proposed heat dissipation structure are not easy to damage under large currents.

Temperature versus current curves of power chips with SF-TTSV array @ *H*_0_ = 600 μm, *H*_1_ = 500 μm and 550 μm, and thinned power chips @ *H*_0_ = 100 μm, 150 μm, 200 μm, 250 μm, and 400 μm, respectively, at 300 K and 398 K ambient temperatures are shown in Figure 11. In order to ensure that the model is completely consistent with other conditions, only the temperature of the heat sink (*T*_hs_) is changed from 300 K to 398 K. Curves in Figure 11 indicate that the heat sink temperature will only increase the overall heat dissipation simulation value of each chip by 98 K and the heat sink temperature will not affect the heat dissipation capacity of power chips, just as the temperature of the radiator increases or decreases as a whole. When the current increases beyond 5 A, the maximum temperature of thinned power chips @ *H*_0_ = 400 μm and *T*_hs_ = 300 K is higher than that of power chips with SF-TTSV array @ *H*_0_ = 600 μm, *H*_1_ = 550 μm and *T*_hs_ = 398 K, and thinned chip @ *H*_0_ = 100 μm and *T*_hs_ = 398 K. The temperature limit of silicon-based power chips is about 473.15 K. Exceeding the temperature limit will lead to a reduction in the life of the device or even direct failure. For thinned power chips @ *H*_0_ = 400 μm and *H*_0_ = 250 μm, *T*_hs_ = 398 K, and the minimum current that causes the chip temperature to exceed the limit temperature is 3.8 A and 5 A, respectively. The temperature of the thinned power chip @ *H*_0_ = 200 μm, *T*_hs_ = 398 K and @ 5.12 A is lower but very close to the temperature limit. Even if temperature curves of thinned power chips @ *H*_0_ = 150 μm and *H*_0_ = 200 μm are close to those of power chips with SF-TTSV @ *H*_0_ = 600 μm, *H*_1_= 500 μm and *H*_1_ = 550 μm, respectively, reducing the thickness of a wafer to less than 200 μm is very difficult to control.

### 3.3. Simulation and Comparative Analysis of Power Chips with Different Current Densities

Different power chips have different application fields, which require that different power chips have different maximum conduction currents and current densities. Under different current densities, the heat dissipation capacity of power chips with the proposed heat dissipation structure and thinned chips are simulated and compared. 

Compared with the simulation conditions in part B, only the current density changed. Temperature curves, under 398 K ambient temperature (*T*_hs_ = 398 K), obtained by varying current densities from 50 to 100 A/cm^2^, of power chips with SF-TTSV array @ *H*_0_ = 600 μm, *H*_1_ = 500 μm and 550 μm, and thinned power chips @ *H*_0_ = 100 μm, 150 μm and 200 μm are shown in Figure 12. For thinned power chips @ *H*_0_ = 200 μm, *H*_0_ = 150 μm and *H*_0_ = 100 μm, the current densities that will cause the temperature to exceed the temperature limit (473.15 K) are about 52.99 A/cm^2^, 61.21 A/cm^2^ and 74.32 A/cm^2^, respectively. For power chips with SF-TTSV array @ *H*_0_ = 600 μm, *H*_1_ = 500 μm and 550 μm, the current densities that will cause the temperature to exceed the temperature limit are about 63.04 A/cm^2^ and 79.04 A/cm^2^, respectively. Compared with thinned power chips @ *H*_0_ = 200 μm and *H*_0_ = 150 μm, power chips with SF-TTSV array @ *H*_0_ = 600 μm and *H*_1_ = 500 μm can conduct higher currents in the same chip area. Even though a thinned power chip @ *H*_0_ = 100 μm has better heat dissipation capability than a power chip with SF-TTSV @ *H*_0_ = 600 μm and *H*_1_ = 500 μm, a power chip with SF-TTSV @ *H*_0_ = 600 μm and *H*_1_ = 550 μm has obviously improved heat conduction capacity than that of a thinned power chip @ *H*_0_ = 100 μm; additionally, the thinner the chip, the more difficult it is to process. 

When the thickness of a chip is 150 μm or less, there are many problems, such as softness, poor rigidity, fragility and easy fracture, that make wafer processing and transmission difficult, and it is easy to cause surface damage and other problems in the process. The proposed heat dissipation structure, in which the SF-TTSV array is embedded into the bulk region of the power chip, is prepared by wet etching without grinding. Maintaining the thickness of the bulk region (*H*_0_) can avoid these problems effectively.

### 3.4. Simulation and Comparative Analysis of SF-TTSV Array Cooling Structure Power Chips with Different Sizes

The proposed heat dissipation structure, in which the SF-TTSV array is embedded into the bulk region of other side-length power chips, and other side-length-thinned power chip structures is simulated under thermoelectric coupling conditions. The chip area is 4 mm × 4 mm and the *H*_0_ of power chips with SF-TTSV array is 600 μm. The thermal resistance and parasitic resistance of different structures can be calculated using the equations in Section 2. The maximum temperatures at certain currents can also be calculated.

The maximum temperatures of power chips with SF-TTSV array @ *H*_1_ = 500 μm and 550 μm, side length = 3.2 mm and 4 mm, respectively, and thinned power chips @ *H*_0_ = 100 μm, 150 μm, 200 μm, 250 μm, side length = 4 mm, respectively, are simulated under different currents. For power chips, the larger the side length, the larger the current cross-sectional area. The large-size power chips have smaller parasitic resistance and thermal resistance, and better heat dissipation capacities. The curves in Figure 12 compare the heat dissipations of different sizes of power chips with the proposed heat dissipation structure under the same current condition and compare the heat dissipation capacity of power chips with an SF-TTSV array and thinned power chips that have the same side length.

As shown in Figure 13, within the 0 A~5.12 A current range, the maximum temperature of the power chip with the proposed heat dissipation structure, whose side length is 4 mm, is lower than the maximum temperature of the power chip with the proposed heat dissipation structure whose side length is 3.2 mm. However, the maximum temperature of a thinned chip @ *H*_0_ = 250 μm and side length = 4 mm is higher than that of a power chip with SF-TTSV array @ *H*_1_ = 550 μm and side length = 3.2 mm; and when this thinned power chip works at 8 A, the maximum temperature of the chip increases from 300 K to 387 K, an increase of 87 K, which means that this thinned power chip cannot work at high ambient temperature and high current environments. For example, if this thinned power chip works in an environment where the ambient temperature is 398 K and the current is 8 A, the maximum temperature of this chip will be about 485 K, which exceeds the temperature limit of the chip, that is 473.15 K, will lead to chip failure or destruction. The curve representing the maximum temperature of the power chip with SF-TTSV array @ *H*_1_ = 500 μm and side length = 4 mm almost completely coincides with the curve representing the maximum temperature of the thinned chip @ *H*_0_ = 150 μm and side length = 4 mm, and the curve representing the maximum temperature of the power chip with SF-TTSV array @ *H*_1_ = 550 μm and side length = 4 mm almost coincides with the curve representing the maximum temperature of the thinned chip @ *H*_0_ = 100 μm and side length = 4 mm.

## 4. Comparative Analysis of the Proposed SF-TTSV and the Cylindrical TSV

For embedded heat dissipation structures using TSVs, cylindrical TSVs are widely used as thermal TSVs. Section 3 shows that the power chip with the proposed heat dissipation structure has better cooling capacity than the thinned power chip; however, the proposed SF-TTSV is not directly comparable with the cylindrical TSV. 

Firstly, the cylindrical TSV is usually prepared using the BOSCH etching process, which is isotropic. It causes scallop patterns on the sides of the TSV wall and may cause reliability problems for the overall system. However, the proposed SF-TTSV was prepared by wet etching on P-type (100) monocrystalline silicon and this method is anisotropic. Note that the proposed SF-TTSV prepared by wet etching has a very smooth side wall, but the side wall of the cylindrical TSV prepared by BOSCH etching has a series of scallop patterns. The scallop patterns would cause many reliability problems for the overall system if no additional process was adopted. Thus, the proposed SF-TTSV prepared by wet etching is better than the cylindrical TSV, which can avoid the reliability problems caused by scallop patterns on the side walls of the TSV. Meanwhile, the process of manufacturing the cylindrical TSV is complex and prone to filling defects. The side walls of SF-TTSVs are inclined and the openings are large, which are conducive to film deposition and copper electroplating filling, which can reduce process difficulty and improve filling quality. Even though the proposed heat dissipation structure based on SF-TTSVs can only be used for the P-type (100) monocrystalline silicon substrate, there are a considerable number of power chips fabricated on the P-type (100) monocrystalline silicon substrate.

Secondly, in the same unit, although the volume of the cylinder is slightly larger than that of the pyramid—which also means that the cylinder will be filled with more copper and the thermal resistance will be smaller than that of the pyramid—the edges and corners of the cylinder are very sharp, and its cross-section is a right angle. Compared with the pyramid, it is more conducive to heat dissipation, which may cause the accumulation of heat and affect the thermal reliability of the system.

Table 3 shows a comparison of the advantages and disadvantages of SF-TTSV, wafer thinning, and cylindrical TSV. From this table, it can be seen that SF-TTSV has advantages of machining accuracy and heat dissipation efficiency compared to wafer thinning; it also has advantages of cost, side wall of deep via, and mechanical reliability compared to cylindrical TSVs. Therefore, embedding the proposed heat dissipation structure based on SF-TTSVs into the silicon substrate of power chips can improve the heat dissipation efficiency of power chips at a relatively low cost.

## 5. Conclusions

In this article, a novel heat dissipation structure in which an SF-TTSV array fabricated by wet etching and refilled with high conductivity and high thermal conductivity materials is proposed and verified by theoretical analysis and simulation based on *COMSOL Multiphysics*. For power chips with P-type (100) silicon substrates, the proposed heat dissipation structure is theoretically analyzed and simulated. Even though CMP is commonly used to reduce the thickness of the silicon substrate to improve heat dissipation efficiency, the process of preparing thinner (<200 μm) chips is more difficult and costly. Theoretical analysis and simulated results indicated that the proposed heat dissipation structure based on SF-TTSVs significantly improved the heat dissipation capacity compared to wafer thinning. Even though the volume of the cylinder is larger than that of the pyramid, the proposed SF-TTSV has advantages during manufacturing and reliabilities compared to cylindrical TSV.

## Figures and Tables

**Figure 1 micromachines-15-00323-f001:**
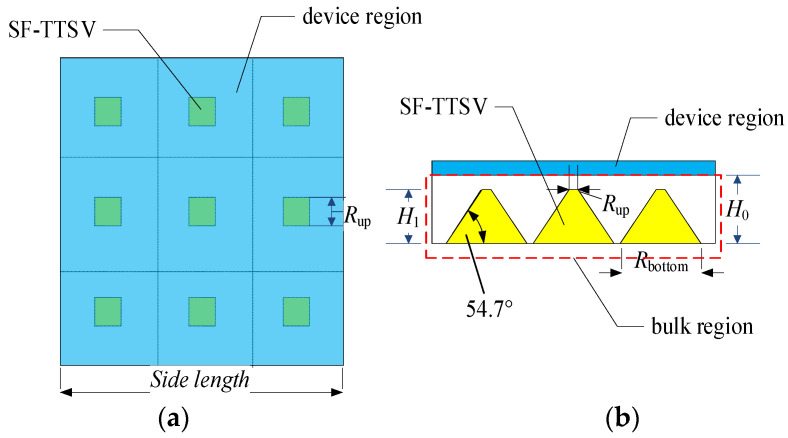
SF-TTSV heat dissipation array: (**a**) Top view; (**b**) Cross-sectional view.

**Figure 2 micromachines-15-00323-f002:**
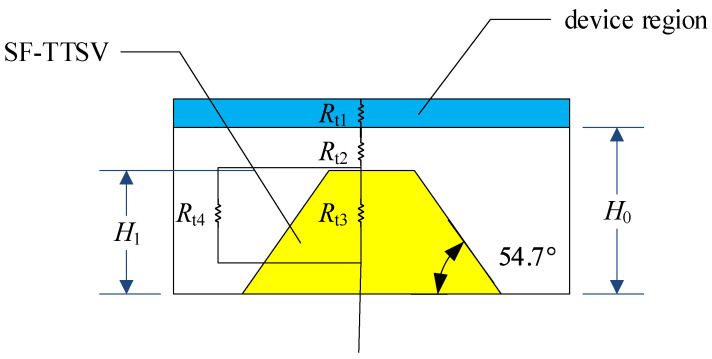
Segmented thermal resistance model of SF-TTSV in the bulk region.

**Figure 3 micromachines-15-00323-f003:**
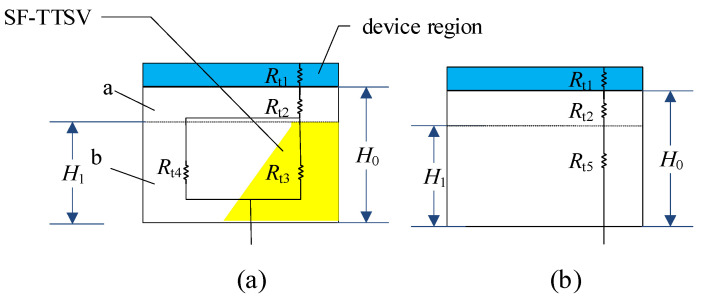
Thermal resistance distribution: (**a**) a cell with half the SF-TTSV. (**b**) the same size cell without SF-TTSV.

**Figure 4 micromachines-15-00323-f004:**
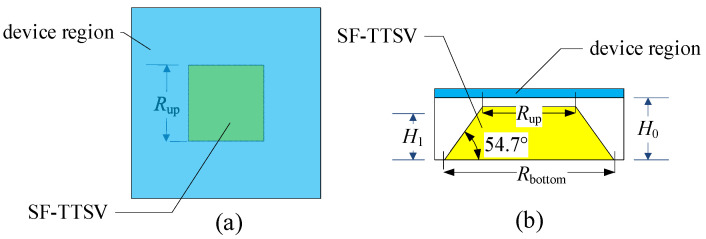
Heat dissipation structure with only one SF-TTSV: (**a**) Top view; (**b**) Cross-sectional view.

**Figure 5 micromachines-15-00323-f005:**
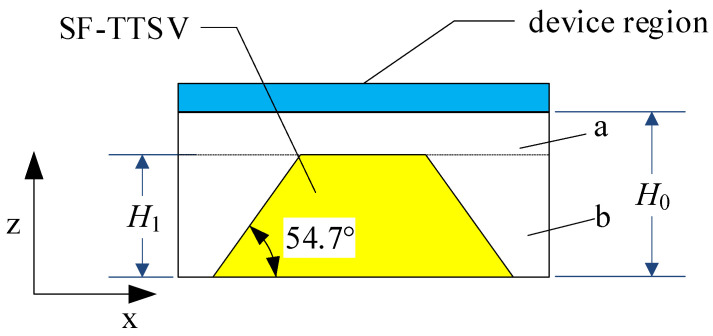
Computation module of z-direction equivalent thermal resistance.

**Figure 6 micromachines-15-00323-f006:**
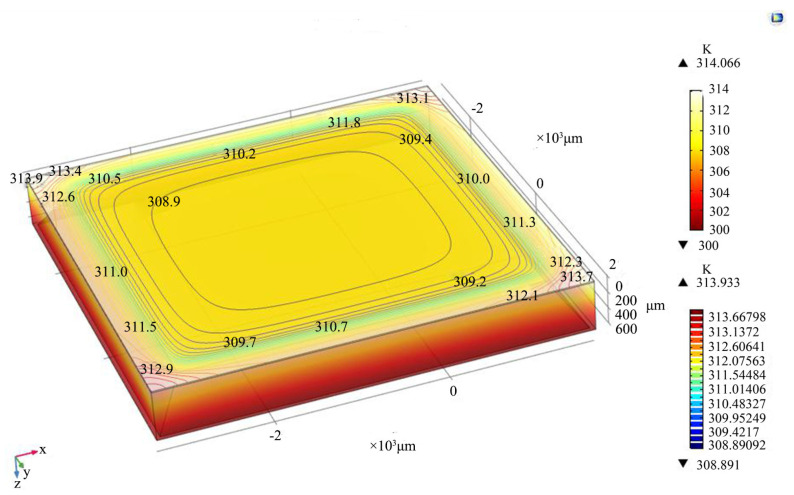
Thermo-electric coupling simulation of a 5 × 5 mm^2^ power chip with one SF-TTSV, @ 10 A current and 300 K ambient temperature.

**Figure 7 micromachines-15-00323-f007:**
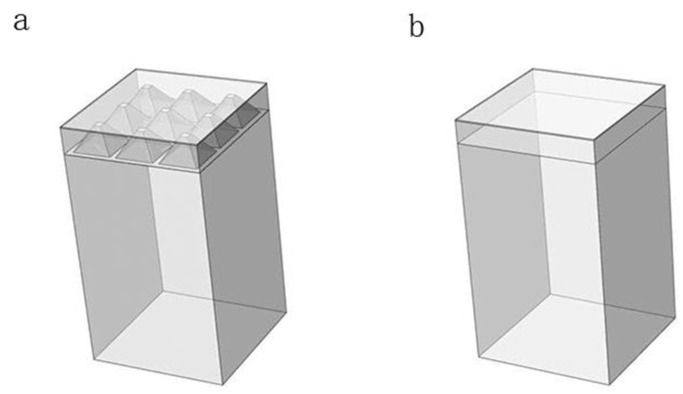
Simulation model of power chips: (**a**) with SF-TTSV array; (**b**) without SF-TTSV.

**Figure 8 micromachines-15-00323-f008:**
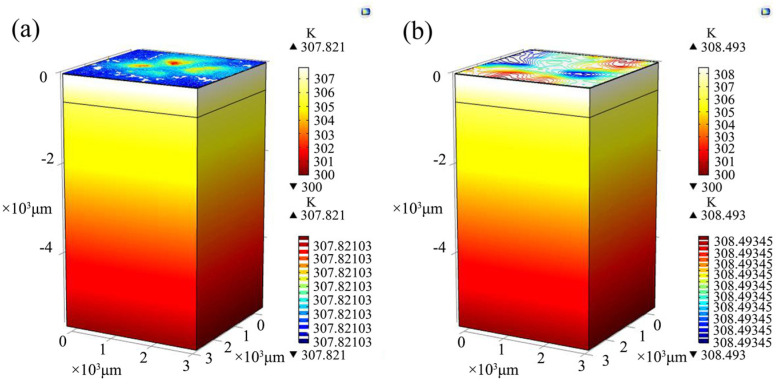
Simulation models of a power chip without SF-TTSV and without thinning: (**a**) heat transfer conditions; (**b**) thermoelectric coupling conditions.

**Figure 9 micromachines-15-00323-f009:**
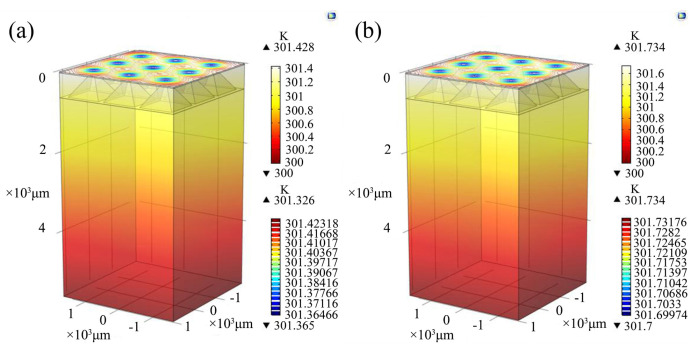
Simulation models of a power chip with an SF-TTSV array: (**a**) heat transfer conditions; (**b**) thermoelectric coupling conditions.

**Figure 10 micromachines-15-00323-f010:**
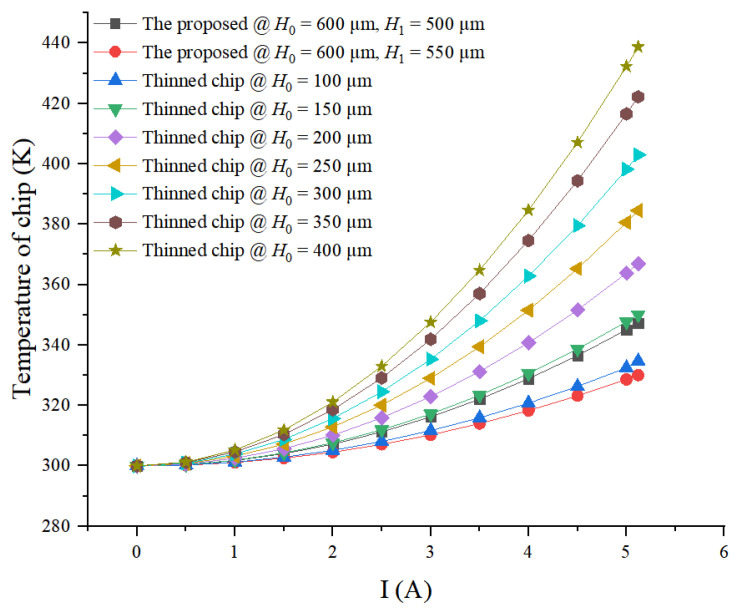
The maximum temperature of power chips with the proposed heat dissipation structure and thinned power chips obtained by varying currents.

**Figure 11 micromachines-15-00323-f011:**
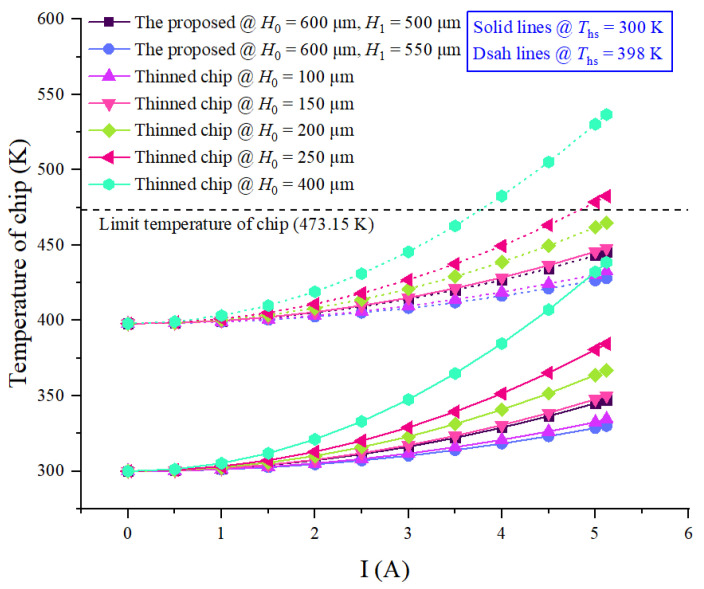
Comparison of the maximum temperatures of power chips with the proposed heat dissipation structure and thinned power chips under different heat sink temperatures.

**Figure 12 micromachines-15-00323-f012:**
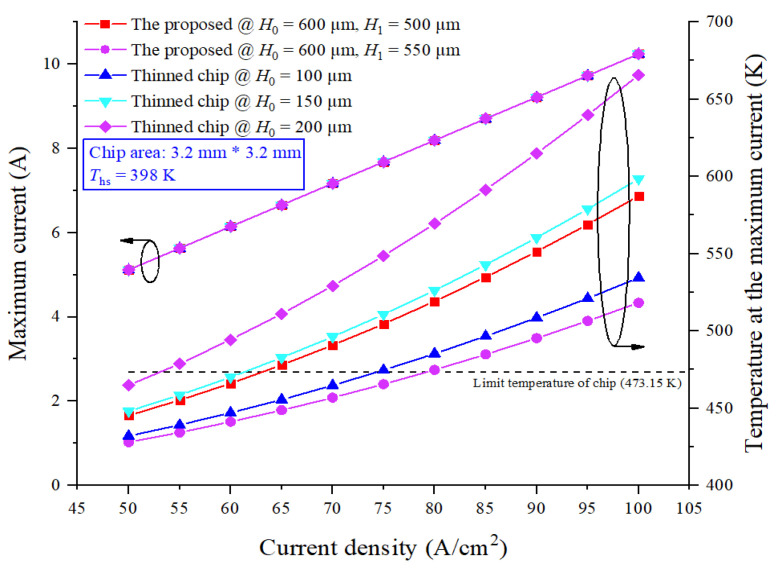
Temperature curves of power chips with the proposed heat dissipation structure and thinned power chips under different current densities.

**Figure 13 micromachines-15-00323-f013:**
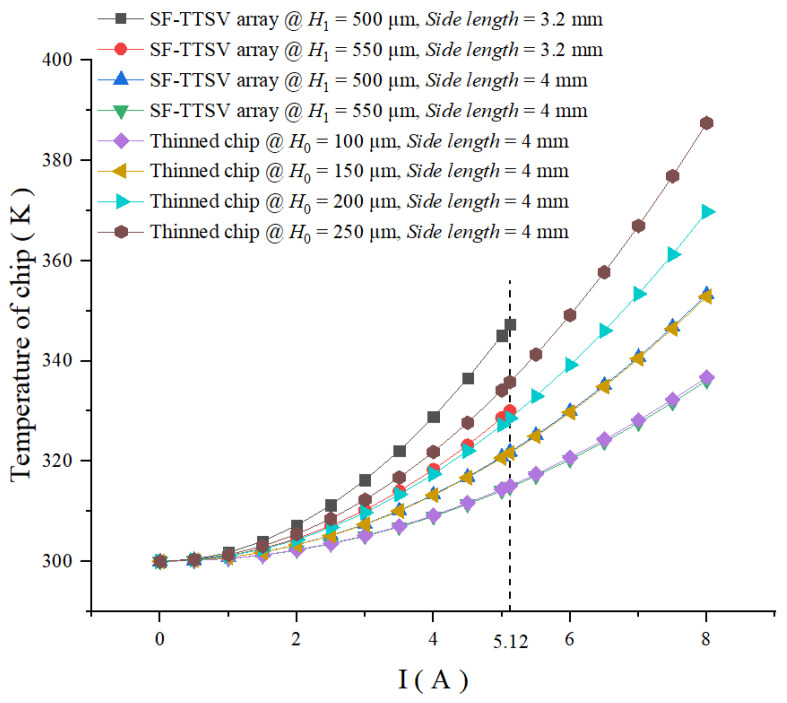
Comparison of maximum temperatures of chips at different side lengths.

**Table 1 micromachines-15-00323-t001:** Parameters for calculation and calculated results.

Parameter	Value	Unit
Volume ratio of SF-TTSV to silicon in part b (*α*)	1.97	---
Equivalent thermal conductivity in the z-direction of part b (*k*_z_)	309.1	W/(m·K)
Total thermal resistance of the bulk region (*R*_t_)	0.1027	K/W
Equivalent conductivity in the z-direction of part b in Figure 5 (*σ*_z_)	39,784,755.89	S/m
Total resistance of the bulk region (*R*_e_)	1	Ω
Dissipated power (*Q*_s_)	100	W
Maximum temperature of the chip (*T*)	312.71	K

**Table 2 micromachines-15-00323-t002:** Different types of chip parameters.

Power Chips	Resistance of the Substrate (mΩ)	Mcc ^1^ (A)
SF-TTSV array *H*_1_ = 500 μm, Side length = 3.2 mm	921.25	5.12
SF-TTSV array *H*_1_ = 550 μm, Side length = 3.2 mm	530.63	5.12
Thinned chip *H*_0_ = 100 μm, Side length = 3.2 mm	921.25	5.12
Thinned chip *H*_0_ = 150 μm, Side length = 3.2 mm	1311.88	5.12
Thinned chip *H*_0_ = 200 μm, Side length = 3.2 mm	1702.50	5.12
Thinned chip *H*_0_ = 250 μm, Side length = 3.2 mm	2093.13	5.12
Thinned chip *H*_0_ = 300 μm, Side length = 3.2 mm	2483.75	5.12
Thinned chip *H*_0_ = 350 μm, Side length = 3.2 mm	2874.38	5.12
Thinned chip *H*_0_ = 400 μm, Side length = 3.2 mm	3265.00	5.12

^1^ Mcc is the maximum conduction current, Jc = 50 A/cm^2^.

**Table 3 micromachines-15-00323-t003:** Comparison of SF-TTSV, wafer thinning and cylindrical TSV.

	SF-TTSV	Wafer Thinning	Cylindrical TSV
Etching technologies	KOH corrosion	CMP	BOSCH etching
Machining accuracy	Low	Higher	High
Cost	Low	Low	High
Side wall of deep via	Smooth	---	Rough
Mechanical reliability	High	High	Low
Silicon substrate	Almost penetrate	≥200 μm	Almost penetrate
Heat dissipation efficiency	High	Low	High

## Data Availability

Data are contained within the article.

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
