# Peer review of "Analysis and Verification of Heat Dissipation Structures Embedded in Substrates in Power Chips Based on Square Frustums Thermal through Silicon Vias"

_micromachines, 2024, doi:10.3390/mi15030323_

Round 1

Reviewer 1 Report

Comments and Suggestions for Authors

The paper describes heat dissipation structure based on square frustum through silicon vias, which can improve heat dissipation capability of power chips. The introduction clearly demonstrates relevance of the work. The heat dissipation structure is analyzed theoretically and by finite element simulation, no experimental data are provided. The paper is interesting for chip design society, particularly for people involved in power electronics. However, the manuscript can be published in Micromachines after some corrections.

1. In the introduction, the authors say “embedded packaging becomes the most advanced ideas recently”. However, existing methods of embedded packaging are not described. I recommend adding some information about known types of through-silicon vias and methods for their formation to the introduction. With this data the paper will be more complete.

2. In lines 67-68, it is said “Compared with thinned chips … the fabrication of the proposed structure is simpler and lower cost”. Can you explain this statement? As far as I know, forming vias in a silicon substrate and filling them with another material requires several technological steps. These operations can be comparable in cost to wafer thinning.

3. The scientific novelty of the work is not clear. Is the pyramidal design of vias proposed for the first time? The novelty should be more clearly highlighted in the manuscript.

4. “The length of heat” in line 108 and “the cross-sectional area of heat” in line 109 sound strange. These expressions need to be reformulated.

5. In Figure 6 there are two temperature scales with different colors. Which one should be used? The same is in Figures 8 and 9.

6. Filling the vias with another material may cause mechanical stress when heated due to different thermal expansion coefficients of silicon and the other material. What materials are suitable for this task? Have you calculated the stress in the wafer during heating?

Comments on the Quality of English Language

Some sentences must be corrected. For example, "This part is do it" in line 422.

Reviewer 2 Report

Comments and Suggestions for Authors

Dear Authors,

Your manuscript looks interesting and important for the research on improving the heat dissipation capacity of power chips while reducing the process difficulty.

The abstract and introduction sections seem relevant to the main text, however, the introduction covers a wide range of topics related to thermal management, including the causes of electronic component failure and various methods to reduce thermal resistance. The presentation could be clearer and more focused. Breaking down the information into smaller, more digestible sections and providing clearer transitions between topics would improve readability. Overall, while the introduction provides a good overview of the importance of thermal management in power semiconductor chips, there is room for improvement to improve the clarity of the presentation. Therefore, the introduction can be improved.

Please consider the following comments:

1. Have the authors performed an uncertainty analysis to assess the robustness of the simulation results? The use of terms such as "error" may be misunderstood by the reader, while you only compare the calculated result and the simulated result. I am sure the uncertainty analysis section is necessary in the manuscript.

2. How do the proposed SF-TTSVs compare to existing heat dissipation techniques in terms of cost effectiveness and scalability?

3. Could the authors elaborate on the fabrication process of the SF-TTSVs, including any challenges encountered and strategies used to overcome them?

4. Are there any limitations or assumptions in the simulation models that could affect the validity of the results?

5. Can the authors provide insight into the potential effects of variations in material properties or geometric parameters on the performance of the SF-TTSV array cooling structure?

6. Has the thermal reliability of the SF-TTSVs been thoroughly evaluated, considering factors such as material fatigue and long-term stability?

7. How does the proposed heat dissipation structure accommodate variations in operating conditions, such as variations in ambient temperature or power load?

8. Are there any practical considerations or trade-offs associated with implementing the SF-TTSV array cooling structure in real-world applications?

9. Can the authors discuss the scalability of the proposed approach for use in different power chip configurations or manufacturing processes?

10. Have the authors considered the environmental impact of the proposed cooling structure, particularly with respect to the choice of materials and manufacturing techniques?

11. The authors use both italic and regular font for equations (e.g. page 5, eq.7, lines 144-145). The reason for this is unclear to me.

12. The title of Table 1 looks wrong.

13. The font on the figures (Fig.6 and others) looks too small, especially on the scale part. Also, the temperature scale (lower one) has too many decimals and looks non-linear. What is the reason for having two temperatures?

Despite the comments made, the authors of the revised paper can be recommended to continue such interesting and important research in this field.

Yours sincerely, the reviewer.

Reviewer 3 Report

Comments and Suggestions for Authors

Dear Authors, interesting article, there are some parts to correct or integrate:

1) Table 1: Correct the table description. Furthermore, enter a brief description of the parameters in the table.

2) In the simulation part with Comsol some information on the parameters used for the simulation is missing. What materials were chosen for the analysis? To better understand the setups I would recommend placing parameter setting tables.

3) There is no numerical analysis, or a given percentage of how much heat dissipation improves. It is necessary to insert a table with the advantages/disadvantages compared to other structures.

4) The conclusions should be written better, highlighting the advantages/disadvantages.

Best Regards

Comments on the Quality of English Language

English is fine

Round 2

Reviewer 1 Report

Comments and Suggestions for Authors

The authors responded to my comments. I recommend acceptance of the paper.

Reviewer 3 Report

Comments and Suggestions for Authors

Dear Authors, thank you for the changes to the paper. They're fine now. No further requests are required.

Best Regards